# The Optimal Tolerance Solution of the Basic Interval Linear Equation and the Explanation of the Lodwick's Anomaly

**Andrzej Piegat** [†] and **Marcin Pluciński** *,[†]

Faculty of Computer Science and Information Systems, West Pomeranian University of Technology, Żołnierska 49, 71-210 Szczecin, Poland; apiegat@wi.zut.edu.pl
* Correspondence: mplucinski@wi.zut.edu.pl
† These authors contributed equally to this work.

**Abstract:** Determining the tolerance solution (TS) of interval linear systems (ILSs) has been a task under consideration for many years. It seems, however, that this task has not been fully and unequivocally solved. This is evidenced by the multiplicity of proposed methods (which sometimes provide different results), the existence of many questions, and the emergence of strange solutions provided by, for example, Lodwick's interval equation anomaly (LIEA). The problem of solving ILEs is probably more difficult than we think. The article presents a new method of ILSs solving, but it is limited to the simplest, basic equation $[\underline{a}, \overline{a}]X = [\underline{b}, \overline{b}]$, which is an element of all more complex forms of ILSs. The method finds the optimal TS for this equation by using multidimensional interval arithmetic (MIA). According to the authors' knowledge, this is a new method and it will allow researchers to solve more complex forms of ILSs and various types of nonlinear interval equations. It can also be used to solve fuzzy linear systems (FLSs). The paper presents several examples of the method applications (including one real-life case).

**Keywords:** interval linear equations; tolerance solution of interval equation; Lodwick anomaly of the interval equation; multidimensional interval arithmetic

## 1. Introduction

The problem presented in this paper fits into a more general class of problems referred to in the research literature as interval-valued optimization problems (IVO-problems) [1,2]. Although the optimization problems have been solved reasonably well for the crisp data case, the quality of their solutions for uncertain data is often not satisfactory. This is evidenced by the large number of studies conducted worldwide. Uncertain data is encountered in all types of systems: in static and dynamic systems, in technical systems (control of moving vehicles, flying or floating objects), and in medical, biological, ecological, economic, and other systems. To be able to satisfactorily solve IVO-problems in these diverse fields, new interval versions of many well-known methods in crisp data optimization, or methods that are completely new and previously unknown, must be developed. The authors of [1,2] give examples of many new scientific tasks, e.g., concepts of convexity, invexity, generalization of convexity, Kuhn–Tucker-pseudoinvexity, control problems with multiple integrals, interval-valued nondifferentiable multi-objective fractional programming problems, vector interval-valued optimization problems with infinite interval constraints and others. All this indicates the enormity of the tasks waiting to be solved by using interval arithmetic.

One such optimization task is the task mentioned in the title of this paper, aimed at finding the tolerance solution of the interval linear equation, which was formulated in the previous century and for which solution methods based on one-dimensional types of interval arithmetic have already been provided [3–7]. As will be shown further on, by applying the new multidimensional interval arithmetic (MIA), it will be possible to solve such tasks, which so far could not be solved. The method presented further also allows us to determine the optimal values of the control variable.

The systems we want to control are characterized by various dependencies determining their operation. One of the simplest systems is a static system realizing the linear relationship $ax = b$, where $a$ is a gain coefficient, $x$ is an input control (decision) variable, and $b$ is an output variable that should have a certain value that we want. In practical tasks, the exact value of the coefficient $a$ is often not known. We only know the interval range of its possible values $[\underline{a}, \overline{a}]$. If we do not know the exact $a$ value, then we cannot guarantee that the output $b$ will be kept at the desired value. The only thing we can do is keep the output $b$ in a certain interval called the tolerance corridor $[\underline{b}, \overline{b}]$. This gives rise to the task of determining one $x$ value or perhaps a set $[\underline{x}, \overline{x}]$ of $x$ values that will "target" the output $b$ in the tolerance corridor. The problem of tolerance control is formulated by the Equation (1).

$$[\underline{a}, \overline{a}]X = [\underline{b}, \overline{b}] \tag{1}$$

The task of solving interval equations (IEs) and determining of the tolerance control (TC) has been dealt with by interval arithmetic for many years [6–10]. A big problem with solving IEs results from the fact that not one but many types of interval arithmetic (IA) exist. The following types of IA are given in [11]: standard IA (SIA), extended (generalized) IA of Kaucher, non-standard (inner) IA of Markov, generalized Hukuhara IA of Dimitrova and Stefanini, optimistic IA of Boukezzoula and Galichet, instantiated IA of Dubois, constrained IA of Lodwick, single-level constrained IA of Chalco-Cano, requisite constrained IA of Klir, gradual IA of Dubois, Prade, Fortin, Boukezzoula, affine IA of Stolfi and Figueredo, and multidimensional IA of Piegat, Pluciński, Landowski.

In general, different IAs can give different results for one and the same problem, which is a paradox. However, new ways of performing interval calculations are still proposed. For example, in [12] there is a proposed IA based on new inverse operations of addition and multiplication and a new concept of the general closed interval. The large number of existing types of IA shows that they are not perfect and that some interval problems cannot yet be solved in a satisfactory and convincing manner. It also proves that the scale of difficulties associated with interval calculations, which at first glance seem very simple, is very large. A number of authors have already written about these difficulties. For example, Kreinovich in [13] draws attention to the need for a deep understanding of the interval problem to be solved and to develop equations that accurately reflect the essence of this problem. Inaccuracies in this task lead to erroneous or partially erroneous calculation results. Dymova in [14] draws attention to the fact that some IAs give different solutions to the problem under consideration, depending on its mathematical form that will be used. This is an unacceptable phenomenon that Mazandarani called "unnatural behavior in modeling" [15]. In the authors' opinion, there are still a number of not fully resolved issues in IA, which should be answered in order to improve this arithmetic. This is all the more important as IA is the basic arithmetic for the fuzzy arithmetic (FA). As Zadeh showed in [16], fuzzy sets (FSs) can be decomposed into $\alpha$-cuts which are intervals. This enables operations of FA to be carried out with the aid of IA.

In order to indicate the theoretical difficulties in contemporary IA, W. Lodwick presented in several of his publications a certain anomaly, which will be hereinafter referred to as Lodwick's interval equation anomaly (LIE-anomaly or LIEA). For the first time, the LIE-anomaly was presented by Lodwick in his keynote-lecture at Congresso Brasileiro de Sistemas Fuzzy, in Sorocaba, Brazil, in 2010. Then it was also described in other publications, e.g., recently in 2017 [17].

*Lodwick's Interval Equation Anomaly*

A system is defined by the relationship $ax = b$ in which we only have an approximate (interval) knowledge of the values $a$ and $b$: $a \in [2, 3]$, $b \in [3, 6]$. If we want to get knowledge about the value of the variable $x$, we have to solve the interval Equation (2).

$$[2, 3]x = [3, 6] \tag{2}$$

This equation can be solved with 2 methods: M1 and M2. The M1 method consists in presenting the Equation (2) in the form of 2 inequalities (3) and (4).

$$[2,3]x \leq [3,6] \quad \rightarrow \quad x \in (-\infty, 1] \tag{3}$$
$$[2,3]x \geq [3,6] \quad \rightarrow \quad x \in [3, \infty) \tag{4}$$

Both of these inequalities yield an empty set of solution (5).

$$x \in (-\infty, 1] \cap [3, \infty) = \varnothing \tag{5}$$

This solution is shown in Figure 1.

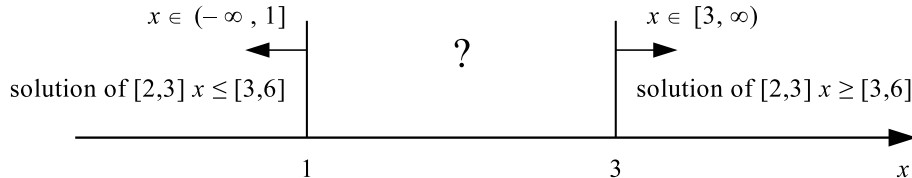

**Figure 1.** Illustration of the solution to the equation $[2,3]x = [3,6]$ with the M1 method which transforms it into two inequalities $[2,3]x \leq [3,6]$ and $[2,3]x \geq [3,6]$.

The solution (5) obtained with the M1 method shows that the analyzed equation has no solution at all. But let's try to solve this equation by using standard interval arithmetic SIA (M2 method). Let's consider, whether there is a proper set $X = [\underline{x}, \overline{x}]$ being a solution to the equation. If it exists, then it must satisfy Equation (6).

$$[2,3][\underline{x}, \overline{x}] = [3,6] \tag{6}$$

The solution of (6) obtained with the use of SIA has the form $[\underline{x}, \overline{x}] = [1.5, 2]$. The correctness of this solution seems to be proved by the fact that it gives equality of the left and right side of Equation (7).

$$[2,3][1.5, 2] = [3,6] \tag{7}$$

Thus, solving Equation (2) with two methods M1 and M2, each of which seems to be completely logical, we obtained two different results (8) and (9), Figure 2.

$$X_1 = [\underline{x}_1, \overline{x}_1] = \varnothing \tag{8}$$
$$X_2 = [\underline{x}_2, \overline{x}_2] = [1.5, 2] \tag{9}$$

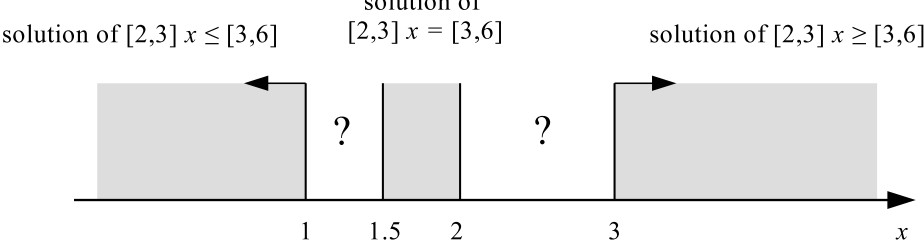

**Figure 2.** Visualization of Lodwick's interval equation anomaly.

The following questions can be asked about LIE-anomalies.

1. Why are the solutions of the equation $[2,3]x = [3,6]$ obtained with the presented methods different?
2. Which of these two methods gives the correct solution? Perhaps, both solutions are correct or both are incorrect?
3. What is the meaning of the mysterious empty intervals $[1, 1.5]$ and $[2, 3]$ shown in Figure 2?

As the LIE anomaly shows, solving interval equations is much more difficult than precise equations. The explanation of this anomaly will be presented in next sections. However, the motivation of this paper is not only to explain the LIE anomaly, but above all to present a new method of determining the tolerance solution (TS) of the basic equation $[\underline{a}, \overline{a}]x = [\underline{b}, \overline{b}]$. Due to the limitation of the volume of the paper, the method of solving the system of many interval equations (IES) will not be presented here. We will cover this in our next article.

The equation $[\underline{a}, \overline{a}]X = [\underline{b}, \overline{b}]$ is the basic element of the IESs studied in the literature since at least 1964 [8–10]. A special case of IESs are the interval linear systems (ILSs). A significant contribution to the development of ILSs was made by S.P. Shary [3–6] in the 1990's. Since then, the terminology used in solving ILSs has been almost unchanged, which can be checked in the latest publications [7,12,17–21].

The mathematical model for the deterministic linear static system is the linear algebraic Equation (10).

$$ax = b \tag{10}$$

In real problems, we can assume that we only know that the coefficients of (10) may independently vary within the intervals $[\underline{a}, \overline{a}] = \mathbf{a}$ and $[\underline{b}, \overline{b}] = \mathbf{b}$ respectively. Thus, we formally have the interval linear algebraic equation.

$$\mathbf{ax} = \mathbf{b} \tag{11}$$

The solution set of (11) can be defined in a variety of ways [6]. We can find the united solution set (12)

$$\sum\nolimits_{\exists\exists}(\mathbf{a}, \mathbf{b}) = \{x \in \mathbb{R} \mid (\exists A \in \mathbf{a})(\exists b \in \mathbf{b})(ax = b)\}, \tag{12}$$

or the *tolerable solution set* formed by all values $x$ such that the product $ax$ falls into $\mathbf{b}$ for any $a \in \mathbf{a}$, i.e., the set (13).

$$\sum\nolimits_{\forall\exists}(\mathbf{a}, \mathbf{b}) = \{x \in \mathbb{R} \mid (\forall A \in \mathbf{a})(\exists b \in \mathbf{b})(ax = b)\} \tag{13}$$

The definitions of different types of ILSs solutions provided by Shary are still valid and are used in subsequent research papers, the authors of which are looking for ILSs solutions based on new ideas, new types of IA [19], and for special cases of ILSs, e.g., two-sided ILS [20,21]. One of the most important articles on ILSs is [7], in which Lodwick and Dubois emphasize the importance of IA for fuzzy linear systems. They present various problems related to IA, and most of all, the problem with some methods providing results that are improper intervals. In [7], they present their own "unified" computation method that will provide solutions in the form of proper or empty intervals but "never" improper ones. The authors of [7], taking into account the existence of various forms of ILSs solutions, propose a model of this system in the form (14):

$$[\mathbf{A}]\tilde{x} \approx [\mathbf{b}], \tag{14}$$

"where $[\mathbf{A}]$ is a matrix whose entries are intervals ... and $\approx$ is a relation between intervals ...". They define the case of the multidimensional tolerance solution set as Shary (15), but they call this set "robust".

$$[\mathbf{A}]x \in [\mathbf{b}] \tag{15}$$

The authors of [7] analyze in detail one-variable interval linear equation, where $[A] = [a] = [\underline{a}, \overline{a}]$, $b = [b] = [\underline{b}, \overline{b}]$, and they define the tolerance solution in the form (16).

$$\Omega_{\forall\exists} = \{x \mid [\underline{a}, \overline{a}]x \subseteq [\underline{b}, \overline{b}]\} \tag{16}$$

The definition (16) is supplemented with the statement that we require

$$\underline{b} \le ax \le \overline{b}, \quad \forall a \in [\underline{a}, \overline{a}]. \tag{17}$$

Lodwick and Dubois analyze in [7] all possible variants of 1V-ILSs with different combinations of signs for $\underline{a}$, $\overline{a}$, $\underline{b}$, $\overline{b}$ and give ready-made solutions for these variants. For example, for 1V-ILS (18):

$$[1,2]x \approx [4,6], \tag{18}$$

with positive values of $\underline{a}$, $\overline{a}$, $\underline{b}$, $\overline{b}$ the tolerance solution set is calculated from Formula (19).

$$\Omega_{\forall\exists} = [\underline{\Omega}_{\forall\exists}, \overline{\Omega}_{\forall\exists}] = \left[ \max_{a \in [1,2]} \min_{b \in [4,6]} \frac{b}{a}, \min_{a \in [1,2]} \max_{b \in [4,6]} \frac{b}{a} \right] = \left[ \max_{a \in [1,2]} \frac{4}{a}, \min_{a \in [1,2]} \frac{6}{a} \right] = [4,3] = \varnothing. \tag{19}$$

Because the calculation method proposed by the authors of [7] gave in this case the result in the form of the improper interval $[4,3]$, it should be assumed that the problem has no solution (the solution set is empty). A similar result is obtained in many tasks of tolerance control where the width of the output state corridor $(\overline{b} - \underline{b})$ is not large enough for the width $(\underline{a}, \overline{a})$. It should also be noted that in all previously cited references, for the 1V-ILS it is assumed that the solution is one-dimensional and that there is an absolute necessity to meet the condition (17): $\underline{b} \le ax \le \overline{b}$, $\forall a \in [\underline{a}, \overline{a}]$. In the opinion of authors of this article, the fulfillment of the condition (17) for $a \in [\underline{a}, \overline{a}]$ for 1V-ILS, is not very realistic. The reasons for this will be presented in Section 2.

The main motivations of this paper are listed below.

- Presentation of the completely new method based on MIA for solving the basic linear interval equation, which is able to solve such equations that cannot be solved with one-dimensional IA.
- Presentation of the method which from the set of all possible solutions of the tolerance control type can indicate the solution that is optimal in terms of control robustness for the uncertainty of system parameters.
- Presentation of the new method of determining tolerance control for $[\underline{a}, \overline{a}]X = [\underline{b}, \overline{b}]$ systems, on the basis of which more advanced versions can be developed for linear interval systems of the second and higher order and for non-linear interval systems.
- Explanation of the meaning of the proposed method of determining tolerance control with the use of problem visualization. One-dimensional IA is not able to solve the problem at hand, as evidenced by Lodwick's anomaly and the examples presented in the paper.
- Presentation of how to determine the optimal tolerance control in the case which is particularly difficult for one-dimensional arithmetic, that is, for the equation $[\underline{a}, 0, \overline{a}]X = [\underline{b}, \overline{b}]$ in which the interval uncertainty on the left side of the equation contains zero.

The rest of the paper is as follows. Section 2 presents uncertainty generators and their influence on solving ILSs. Section 3 presents a method of determining realistic solutions of the basic equation $[\underline{a}, \overline{a}]x = [\underline{b}, \overline{b}]$ for the case of positive values of $\underline{a}$, $\overline{a}$, $\underline{b}$, $\overline{b}$. Section 4 will consider the cases of intervals containing zeros, that is $[\underline{a}, 0, \overline{a}]$, $[\underline{b}, 0, \overline{b}]$. Section 5 contains conclusions. To the authors' knowledge, the proposed method of a realistic solution of interval equations is new.

## 2. Uncertainty Generators and Their Influence on Solving ILSs

In the system determined by the equation $[a]x = [b]$, the uncertainty of the coefficient $a \in [\underline{a}, \overline{a}]$ adversely affects the quality of control and the possibility of "hits" in the tolerance corridor $[\underline{b}, \overline{b}]$. It should be added here that the uncertainty $[\underline{b}, \overline{b}]$ determining the control tolerance is not a control disturbance (difficulty), but facilitates the control objective. If, with one tolerance $[\underline{b}, \overline{b}]$, it is not possible to find a tolerance solution, then maybe it will be found after increasing the tolerance. However, in practice, such an enlargement is often

not possible due to precision requirements. As shown by the authors of [6,7], as well as other publications, with just one disturbance uncertainty generator (DUG), it is quite often impossible to implement TC due to obtaining empty sets of solutions. Note, however, that in the 2V-ILS system there are as many as 4 DUGs and only 2 tolerance uncertainties (TUs) $[\underline{b}_1, \overline{b}_1]$, $[\underline{b}_2, \overline{b}_2]$, (20).

$$[\underline{a}_{11}, \overline{a}_{11}]x_1 + [\underline{a}_{12}, \overline{a}_{12}]x_2 = [\underline{b}_1, \overline{b}_1] \qquad (20)$$
$$[\underline{a}_{21}, \overline{a}_{21}]x_1 + [\underline{a}_{22}, \overline{a}_{22}]x_2 = [\underline{b}_2, \overline{b}_2]$$

It is understandable that in a 2V-ILS system, it will be much more difficult to implement TC. Let us now consider the uncertain quadratic system (21).

$$[\underline{a}_2, \overline{a}_2]x^2 + [\underline{a}_1, \overline{a}_1]x + [\underline{a}_0, \overline{a}_0] = [\underline{b}, \overline{b}] \qquad (21)$$

In this system, there is only one TC $[\underline{b}, \overline{b}]$ per 3 DUGs. This is a worse situation than in the system (20). Additionally, complex solutions may appear in the quadratic system, which will make it even more difficult to obtain TC. If we want to implement TC in the 3V-ILS system, the number of DUGs is then 9 and the number of TUs is 3. It can be seen that in $n$V-ILS systems the number of DUGs grows much faster ($n^2$) than the number of TUs, which is $n$. Therefore, in more complex systems, the implementation of TC becomes practically impossible. An example here is the task of hitting a plane with a missile fired from an anti-aircraft gun. The following DUGs appear here: uncertainty in measuring the $x$, $y$, $z$ position, uncertainty in measuring the vertical and horizontal angle of the plane's movement direction, uncertainty in measuring the plane's speed, uncertainty of the initial velocity of the fired projectile, error in setting the horizontal and vertical angle of the cannon barrel, uncertainty of wind speed and direction, and other. Is it possible to guarantee a hit in the silhouette of the plane with such a number of DUGs? This is a rather unrealistic task. Therefore, the authors believe that the realistic goal of TC is not to look for the value of the control (decision) variable, which guarantees reliable implementation of TC, but to look for its value that will give the maximum probability of TC implementation—and if possible, full probability, i.e., certainty. In the next section, we will briefly introduce the multidimensional interval arithmetic (MIA) that will be used to realize realistic TC.

### 3. A Brief Introduction to Multidimensional Interval Arithmetic

The MIA concept was developed in 2010–2011, and the first publication [22] on this subject appeared in 2012. Then other publications appeared, e.g., [23,24]. The authors of MIA used this arithmetic to create multidimensional fuzzy arithmetic (MFA Type 1) by using the $\alpha$-cuts principle [25]. In turn, the MFA Type 1 was used to develop the MFA Type 2, e.g., [26–28]. The research team of A. Piegat, M. Pluciński, M. Landowski and others by the beginning of 2022 published 47 articles on MIA, MFA Type 1 and 2. MFA has met with great interest from many scientists who have applied it to solve various problems. By the beginning of 2022, more than 40 application publications had been published, e.g., [29,30].

The main feature that distinguishes MIA from all other types of IA is the form of the computed result. In MIA, the result of an arithmetic operation on intervals is not an interval, i.e., a one-dimensional mathematical object, but is a multidimensional result (set) depending on the number of intervals involved in the operation. This is important in case of complex problems and prevents the loss of information during the computation. In MIA, the interval $[\underline{a}, \overline{a}]$ is transformed into the form of relative-distance-measure (RDM), (22).

$$[\underline{a}, \overline{a}] \rightarrow \underline{a} + \gamma_a(\overline{a} - \underline{a}), \quad \gamma_a \in [0, 1] \qquad (22)$$

The model (22) is epistemic [7], i.e., it is a model of a single, true value of $a$-variable, which, however, we do not know.

The method of performing the interval addition operation is presented in (23).

$$a(\gamma_a) + b(\gamma_b) = c(\gamma_a, \gamma_b), \quad \gamma_a, \gamma_b \in [0, 1] \tag{23}$$
$$[\underline{a} + \gamma_a(\overline{a} - \underline{a})] + [\underline{b} + \gamma_b(\overline{b} - \underline{b})] = (\underline{a} + \underline{b}) + \gamma_a(\overline{a} - \underline{a}) + \gamma_b(\overline{b} - \underline{b})$$

The interpretation of the addition operation is as follows: we know the real value of neither the variable $a$ nor $b$, therefore it is impossible to determine the exact value of the result $c = a + b$. What can we do in this situation? We can only, on the basis of our knowledge, define a set of possible conditional hypotheses. For example, if $(a = 2)$ and $(b = 3)$ then $(c = 5)$. Each of the hypotheses can be presented generally in the form of triple (24).

$$(a(\gamma_a), b(\gamma_b), c(\gamma_a, \gamma_b)) = a(\gamma_a) + b(\gamma_b)) \tag{24}$$

If $[\underline{a}, \overline{a}] = [1, 2]$ and $[\underline{b}, \overline{b}] = [3, 5]$ then examples of possible hypotheses are as follows:

$$(1.00, 3.00, 4.00), (1.01, 3.01, 4.02), \ldots. \tag{25}$$

The set $S_{A+B}$ of all possible solutions $\{(a(\gamma_a), b(\gamma_b), c(\gamma_a, \gamma_b))\}$ is a 3-dimensional set and the result $c(\gamma_a, \gamma_b)$ depends on $a(\gamma_a)$ and $b(\gamma_b)$. Figure 3 illustrates the operation of adding 2 interval sets $[a]$ and $[b]$ containing all possible values of these variables and shows the set of possible results of adding $c(\gamma_a, \gamma_b)$ in 2D-space $A \times B$.

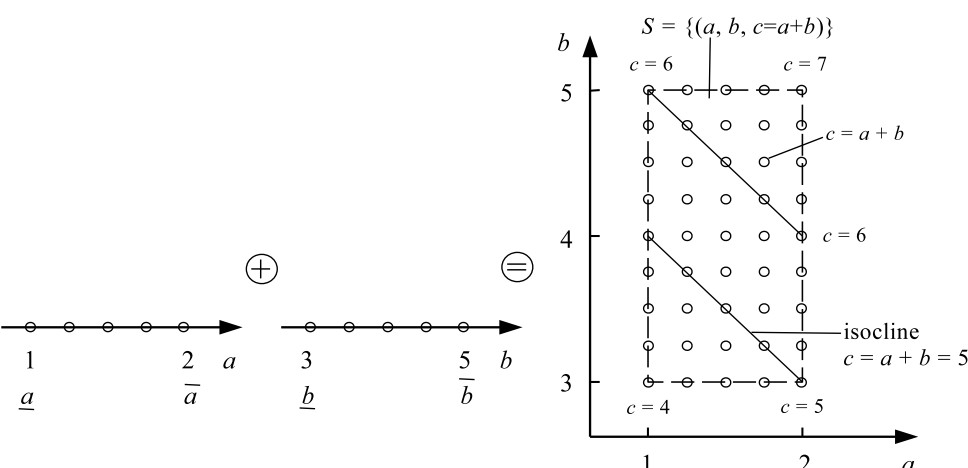

**Figure 3.** Illustration of the operation of adding 2 interval sets $[a]$ and $[b]$ and the 3D result of this operation $S\{(a, b, c)\}$ presented as a projection on 2D-space $A \times B$.

The 3D-result of adding $S_{A+B} = \{(a, b, c)\}$ is the set of all hypothetical states of the addition system. In this set, we can distinguish many states $(a, b, c)$ containing the same result $c$ with different values of $a$ and $b$, e.g., $(1, 4, 5)$, $(1.5, 3.5, 5)$, $(2, 3, 5)$, etc. This means that all the $c$ values in the set $S_{A+B}$ form a bag (they contain repeating values). The bag $BG_c$ can be defined by Formula (26).

$$BG_C = \{c(\gamma_a, \gamma_b) = a(\gamma_a) + b(\gamma_b) \mid \forall \gamma_a \in [0, 1], \forall \gamma_b \in [0, 1]\} \tag{26}$$

However, for practical calculations the span of the bag $SP_{BG_C}$ will be used (27). The span does not contain the same repeated values of $c$.

$$SP_{BG_C} = \left[\min_{\gamma_a, \gamma_b} c(\gamma_a, \gamma_b), \max_{\gamma_a, \gamma_b} c(\gamma_a, \gamma_b)\right], \quad \gamma_a, \gamma_b \in [0, 1] \tag{27}$$

In the case of intervals $[\underline{a}, \overline{a}] = [1, 2]$ and $[\underline{b}, \overline{b}] = [3, 5]$, their RDM form is given by (28).

$$[1,2] \to a(\gamma_a) = 1 + \gamma_a, \quad \gamma_a \in [0,1] \tag{28}$$
$$[3,5] \to b(\gamma_b) = 3 + 2\gamma_b, \quad \gamma_b \in [0,1]$$
$$a(\gamma_a) + b(\gamma_b) = c(\gamma_a, \gamma_b) = 4 + \gamma_a + 2\gamma_b$$

Based on (27), we calculate the span $SP_{BG_C}$ given by (29).

$$SP_{BG_C} = [4,7] \tag{29}$$

The meaning of the span can be seen in Figure 3. This is the span of the set $S_{A+B}$ in the direction of $c$. The second simplified information about the set of possible states $S_{A+B}$ is the normalized distribution of the cardinality measure $CardM(c)$ that informs about the number of possibilities in which a particular result $c$ can occur (see Figure 4). $CardM(c)$ can be derived from the length of isoclines $c = a + b = const$, as shown in Figure 3.

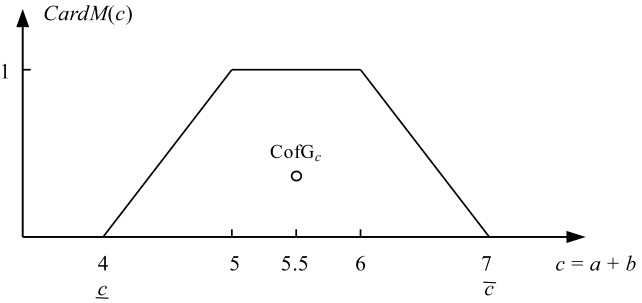

**Figure 4.** Normalized distribution of the cardinality measure $CardM(c)$ of the set of possible result values $c = a + b$ in the interval addition operation $[1,2] + [3,5]$.

From the distribution $CardM(c)$, it is possible to calculate the center of gravity $CofG_C$ of the bag $BG$ being another simplified information about the result variable $c$.

$$CofG_C = \frac{\int_{\underline{c}}^{\overline{c}} c \cdot CardM(c)\, \mathrm{d}c}{\int_{\underline{c}}^{\overline{c}} CardM(c)\, \mathrm{d}c} = 5.5 \tag{30}$$

The above description shows that the result of adding the intervals $[a] + [b]$ is not unequivocal. There is a multivariate result set $S_{A+B} = \{(a(\gamma_a), b(\gamma_b), c(\gamma_a, \gamma_b) = a(\gamma_a) + b(\gamma_b))\}$ and secondary simplified results: span $SP_{BG_C}$, 2D-distribution $CardM(c)$ and center of gravity $CofG_c$. It is similar to other arithmetic operations. For each of them, the main result set and secondary results from this set can be determined.

*3.1. Subtraction of Proper Intervals [a] − [b]*

Result variable $c(\gamma_a, \gamma_b)$:

$$c(\gamma_a, \gamma_b) = a(\gamma_a) - b(\gamma_b), \quad \gamma_a, \gamma_b \in [0,1], \tag{31}$$
$$[\underline{a} + \gamma_a(\overline{a} - \underline{a})] - [\underline{b} + \gamma_b(\overline{b} - \underline{b})] = (\underline{a} - \underline{b}) + \gamma_a(\overline{a} - \underline{a}) - \gamma_b(\overline{b} - \underline{b}).$$

The resulting set $S_{A-B}$ of states of the subtraction system is given by (32).

$$S_{A-B} = \{(a(\gamma_a), b(\gamma_b), c(\gamma_a, \gamma_b) = a(\gamma_a) - b(\gamma_b)) \mid \forall \gamma_a \in [0,1],\ \forall \gamma_b \in [0,1]\} \tag{32}$$

*3.2. Multiplication of Proper Intervals [a][b]*

Result variable $c(\gamma_a, \gamma_b)$:

$$c(\gamma_a, \gamma_b) = a(\gamma_a)b(\gamma_b), \quad \gamma_a, \gamma_b \in [0,1]. \tag{33}$$

Resulting set $S_{AB}$ of states:

$$S_{AB} = \{(a(\gamma_a), b(\gamma_b), c(\gamma_a, \gamma_b) = a(\gamma_a)b(\gamma_b)) \mid \forall \gamma_a \in [0,1],\ \forall \gamma_b \in [0,1]\}. \tag{34}$$

### 3.3. Dividing of Proper Intervals [a] / [b]

Result variable $c(\gamma_a, \gamma_b)$:

$$c(\gamma_a, \gamma_b) = a(\gamma_a)/b(\gamma_b), \quad 0 \notin b(\gamma_b), \ \gamma_a, \gamma_b \in [0,1]. \tag{35}$$

Resulting set $S_{A/B}$ of states:

$$S_{A/B} = \{(a(\gamma_a), b(\gamma_b), c(\gamma_a, \gamma_b) = a(\gamma_a)/b(\gamma_b)) \mid \forall \gamma_a \in [0,1], \ \forall \gamma_b \in [0,1]\}. \tag{36}$$

If $0 \in [b]$, which can be represented as $[\underline{b}, 0, \overline{b}]$ where $\underline{b} < 0$, $\overline{b} > 0$, then in order to perform a division by $[b]$ the operation can be transformed into an approximate form (37) containing a very small positive number $\Delta$, e.g., $\Delta = 0.001$.

$$[\underline{b}, 0, \overline{b}] = [\underline{b}, -\Delta] \cup [\Delta, \overline{b}] \tag{37}$$

The entire division operation can be decomposed into a union of two-component division operations, as shown in (38).

$$[\underline{a}, \overline{a}] / [\underline{b}, 0, \overline{b}] = [\underline{a}, \overline{a}] / [\underline{b}, -\Delta] \cup [\underline{a}, \overline{a}] / [\Delta, \overline{b}] \tag{38}$$

## 4. Method of Determining Realistic Tolerance Solutions

A static multiplicative system is given with inputs $a$ and $x$ and with output $b$ realizing the operation $ax = b$. We have an approximate knowledge about the true value $a$: $a \in [\underline{a}, \overline{a}]$, $\underline{a} \le \overline{a}$, and about $b$: $b \in [\underline{b}, \overline{b}]$, $\underline{b} \le \overline{b}$. We have no knowledge about the value of $x$ and we want to gain this knowledge. In this situation, the variables $a$ and $b$ become information inputs and $x$ information output. The knowledge we can get about $x$ depends on the knowledge about $a$ and $b$. We can present the whole problem in the form (39).

$$ax = b, \ a \in [\underline{a}, \overline{a}], \ b \in [\underline{b}, \overline{b}], \ x = ? \tag{39}$$
$$a \to a(\gamma_a) = \underline{a} + \gamma_a(\overline{a} - \underline{a}), \ \gamma_a \in [0,1]$$
$$b \to b(\gamma_b) = \underline{b} + \gamma_b(\overline{b} - \underline{b}), \ \gamma_b \in [0,1]$$
$$ax = b \to a(\gamma_a)x(\gamma_a, \gamma_b) = b(\gamma_b)$$

With the use of MIA, it is possible to determine the knowledge about the output $x$, (40).

$$x(\gamma_a, \gamma_b) = \frac{b(\gamma_b)}{a(\gamma_a)} = \frac{\underline{b} + \gamma_b(\overline{b} - \underline{b})}{\underline{a} + \gamma_a(\overline{a} - \underline{a})}, \ \gamma_a, \gamma_b \in [0,1] \tag{40}$$

For one selected numerical $\gamma_a$ and $\gamma_b$ value, Formula (40) determines one value of $a(\gamma_a)$ and $b(\gamma_b)$ and one value $x(\gamma_a, \gamma_b)$ associated with them. In this sense, Formula (40) is a mathematical model of the true values of $a$, $b$ and $x$. However, for all values of $\gamma_a \in [0,1]$ and $\gamma_b \in [0,1]$, this formula determines $BG_X$, that is the bag of all possible values of $x$ corresponding to all combinations of values $\gamma_a$ and $\gamma_b$, Formula (41).

$$BG_X = \{x(\gamma_a, \gamma_b) = b(\gamma_b)/a(\gamma_a) \mid \forall \gamma_a \in [0,1], \forall \gamma_b \in [0,1]\} \tag{41}$$

Given the formula of $BG_X$, we can determine the set $X = [\underline{x}, \overline{x}]$ of all possible values of $x$, which does not contain the repetitions of identical values, (42).

$$SP_{BG_X} = X = [\underline{x}, \overline{x}] = \left[ \min_{\gamma_a, \gamma_b} x(\gamma_a, \gamma_b), \max_{\gamma_a, \gamma_b} x(\gamma_a, \gamma_b) \right], \ \gamma_a, \gamma_b \in [0,1] \tag{42}$$

Because the set $X$ is a set in 1D-space, it can be represented in RDM terms as (43).

$$X = X_{poss} = \{x(\gamma_x) = \underline{x} + \gamma_x(\overline{x} - \underline{x}) \mid \forall \gamma_x \in [0,1]\} \tag{43}$$

In general, the set $X_{poss}$ consists of 2 subsets: $X_{FR}$ – set of fully robust (adjusted) $x$-values) and $X_{PR}$ – set of $x$-values that are only partly robust, partly adjusted to the disturbance $a \in [\underline{a}, \overline{a}]$. Both subsets are expressed by (44).

$$X_{poss} = X_{FR} \cup X_{PR} \tag{44}$$

$$X_{FR} = \{x \mid \forall a \in [\underline{a}, \overline{a}] : \ ax = b \in [\underline{b}, \overline{b}]\}$$

$$X_{PR} = \{x \mid (\exists a \in [\underline{a}, \overline{a}] : \ ax = b \in [\underline{b}, \overline{b}]) \text{ and } (\exists a \in [\underline{a}, \overline{a}] : \ ax = b \notin [\underline{b}, \overline{b}])\}$$

The authors' research shows that $X_{FR}$ sets are rather rare in real problems and often only $X_{PR}$ sets occur. Now, the algorithm of solving the tolerance control problem will be given.

Step 1: Formulate the interval sets $[\underline{a}, \overline{a}]$ and $[\underline{b}, \overline{b}]$ in terms of MIA.
Step 2: Determine the bag model $BG_X = \{x(\gamma_a, \gamma_b) \mid \forall \gamma_a \in [0,1], \forall \gamma_b \in [0,1]\}$.
Step 3: Specify bag span $SP_{BG_X} = X_{poss} = [\underline{x}, \overline{x}]$.
Step 4: Determine the degrees of robustness $r(x)$ for the individual $x$-values of the $X_{poss}$ set to the disturbance $A = [\underline{a}, \overline{a}]$.
Step 5: Determine the optimal tolerance control $x_{opt}$ value based on the selected decision evaluation criterion.

The degree of robustness $r(x^*)$ of the chosen control value $x^*$ for the range $[\underline{a}, \overline{a}]$ of possible values of the uncertain disturbance $a$ is a fraction ($r(x^*) \in [0,1]$) informing how large part of the range $[\underline{a}, \overline{a}]$ for the value $x^*$ will give the product $ax^*$ value in the required tolerance corridor $[\underline{b}, \overline{b}]$. The robustness $r(x^*)$ is simply the ratio of lengths of two segments marked in Figure 5.

To determine the robustness degree, follow the steps.

Step A: For the $x^*$ value, calculate the upper $a^U(x^*) = \overline{b}/x^*$ and the lower $a^L(x^*) = \underline{b}/x^*$ coordinates of the points on the upper $ax = \overline{b}$ and lower $ax = \underline{b}$ border of the tolerance corridor $[\underline{b}, \overline{b}]$.

Step B: Determine the common part of the intervals $[a^L(x^*), a^U(x^*)]$ and $[\underline{a}, \overline{a}]$:

$$[a_c(x^*)] = [a^L(x^*), a^U(x^*)] \cap [\underline{a}, \overline{a}] = [a_c^L(x^*), a_c^U(x^*)]. \tag{45}$$

Step C: Calculate the robustness degree for the considered value of $x^*$:

$$r(x^*) = \frac{a_c^U(x^*) - a_c^L(x^*)}{\overline{a} - \underline{a}}. \tag{46}$$

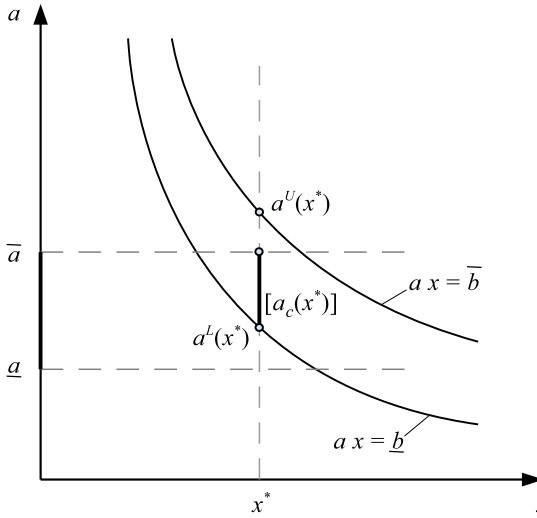

**Figure 5.** Illustration of the computation method of $r(x^*)$.

In the following, the above algorithm will be applied to the solution of the example of Lodwick and Jenkins in the invited talk "Uncertainties in mathematical analysis and their use in optimization" presented at Congresso Brasileiro de Sistemas Fuzzy in Sorocaba, Brasil, 2010.

**Example 1.** *Let's consider a system with inputs a and x and the output b realizing the relationship $ax = b$. We have the following knowledge about the real values of the input a : $a \in [\underline{a}, \overline{a}] = [2, 3]$. The input a is a generator of uncertainty (disturbance of the control process), which we have no influence on. The output b is the control target for which a realistic, tolerant requirement $b \in [\underline{b}, \overline{b}] = [3, 6]$ was made. Tolerance $[\underline{b}, \overline{b}]$ is not a disturbance, but a facilitation of the control process. The task here is to determine one, or perhaps a set of control values x, which will allow the control objective to be achieved in the best possible way. In summary, our knowledge of the system is given by (47).*

$$ax = b, \quad a \in [\underline{a}, \overline{a}] = [2, 3], \quad b \in [\underline{b}, \overline{b}] = [3, 6], \quad x =? \tag{47}$$

*Step 1.*

$$[\underline{a}, \overline{a}] = [2, 3] \rightarrow a(\gamma_a) = \underline{a} + \gamma_a(\overline{a} - \underline{a}) = 2 + \gamma_a, \; \gamma_a \in [0, 1] \tag{48}$$

$$[\underline{b}, \overline{b}] = [3, 6] \rightarrow b(\gamma_b) = \underline{b} + \gamma_b(\overline{b} - \underline{b}) = 3 + 3\gamma_b, \; \gamma_b \in [0, 1]$$

*Step 2.*

$$ax = b \rightarrow x(\gamma_a, \gamma_b) = \frac{b(\gamma_b)}{a(\gamma_a)} = \frac{3 + 3\gamma_b}{2 + \gamma_a}, \quad \gamma_a, \gamma_b \in [0, 1] \tag{49}$$

$$BG_X = \left\{ x(\gamma_a, \gamma_b) = \frac{3 + 3\gamma_b}{2 + \gamma_a} \; | \; \forall \gamma_a \in [0, 1], \forall \gamma_b \in [0, 1] \right\}$$

*Step 3.*

$$SP_{BG_X} = \left[ \min_{\gamma_a, \gamma_b} \frac{3 + 3\gamma_b}{2 + \gamma_a}, \max_{\gamma_a, \gamma_b} \frac{3 + 3\gamma_b}{2 + \gamma_a} \right] = [1, 3] = [\underline{x}, \overline{x}] = X_{poss} \tag{50}$$

*A minimum of $x(\gamma_a, \gamma_b)$ is obtained for $\gamma_a = 1$, $\gamma_b = 0$ and a maximum of $x(\gamma_a, \gamma_b)$ for $\gamma_a = 0$, $\gamma_b = 1$. Figure 6 illustrates the meaning of the bag $BG_X$ of possible system states ($a, x = b/a, b$) satisfying the control objective, and the meaning of the span $SP_{BG_X} = X_{poss} = [\underline{x}, \overline{x}]$.*

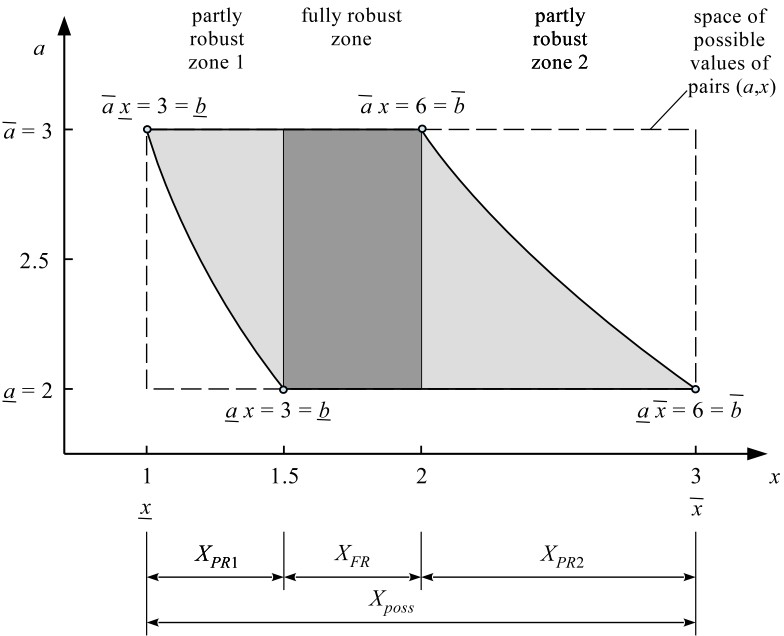

**Figure 6.** Illustration of solution zones (tolerant control zones) in the Lodwick anomaly problem (47).

As shown in Figure 6, the selection of the control value $x \in X_{FR} = [1.5, 2]$ provides fully robust tolerance control, because regardless of what value the disturbance $a$ from the range $[\underline{a}, \overline{a}] = [2, 3]$ takes, the output $b$ will always be included in the desired range $[\underline{b}, \overline{b}] = [3, 6]$. If we select the control value $x$ in the partly robust zone 1, $X_{PR1} = [1, 1.5]$, there is no certainty that $b$ will be included in the tolerance zone $[3, 6]$. For example, if we choose $x = 1.2$, then the output $b$ will be included in $[3, 6]$ only when the disturbance $a$ is included in the range $[2.5, 3]$. If $a$ is in the range $[2, 2.5]$ then the output $b < \underline{b} = 3$. Hence the value $x = 1.2$ has the robustness $r(1.2) = 0.5$. It means that for this $x$-value one half of the disturbance range $a \in [2.5, 3]$ results in hitting the value $ax$ in the tolerance corridor $[\underline{b}, \overline{b}] = [3, 6]$. On the other hand, the second half $[2, 2.5]$ excludes such a hit. If the true disturbance value is in the range $[2.5, 3]$, the hit will occur, if it is in the range $[2, 2.5]$, there will be no hit and the control objective will not be met.

The robustness degree $r(x)$ of the selected $x$-value in zone 1 can be determined from Formula (51).

$$r(x) = \frac{\overline{a} - a(x)}{\overline{a} - \underline{a}} = \frac{\overline{a} - (\underline{b}/x)}{\overline{a} - \underline{a}} \tag{51}$$

It is easy to see that for zone 1, $r(x) \in [0, 1]$, has a fractional value denoting incomplete robustness for the disturbance $a$. In the $X_{FR} = [1.5, 2]$ zone, every control value $x$ is completely robust ($r(x) = 1$) for all possible disturbance values $a \in [2, 3]$ providing $b \in [3, 6]$. In the partly robust zone 2, $X_{PR2} = [2, 3]$ we have a partial robustness $r(x)$ for the individual values of $x$, as seen in Formula (52).

$$r(x) = \frac{a(x) - \underline{a}}{\overline{a} - \underline{a}} = \frac{(\overline{b}/x) - \underline{a}}{\overline{a} - \underline{a}} \tag{52}$$

As shown by the MIA solution of the interval equation $[\underline{a}, \overline{a}]x = [\underline{b}, \overline{b}]$ considered by Lodwick, there are no anomalies in this equation if it is solved by using the algorithm presented in this section. The middle zone $x \in [1, 3]$ shown in Figure 7 does not contain any empty and mysterious sub-zones. It comprises 2 zones of partial robustness and 1 zone of full robustness to disturbance (uncertainty) $a$. The left outer zone $x \leq 1$ is the zone of completely no tolerance control, as is the right zone $x \geq 3$. Figure 6 showing the distribution of the robustness degree $r(x)$ explains Lodwick's interval anomaly.

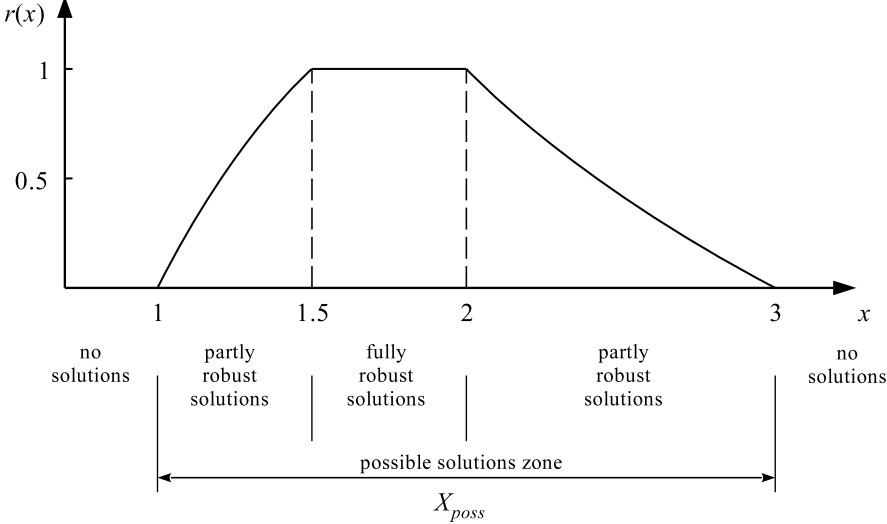

**Figure 7.** Distribution of the robustness degree $r(x)$ of control $x$ as illustration explaining Lodwick's interval equation anomaly.

The discovery of Lodwick's anomaly was caused by the fact that the studied equation $[2, 3]X = [3, 6]$ was solved with the commonly used assumption that the solution $X$ of

this equation is the interval $[\underline{x}, \overline{x}]$. In other words, the equation $[2, 3][\underline{x}, \overline{x}] = [3, 6]$ was solved, which gave the result $[\underline{x}, \overline{x}] = [1.5, 2]$. Such an assumption is incorrect and for many interval equations it does not provide generally feasible solutions in the form of proper intervals. For example, in the case of the equation $[1, 3][\underline{x}, \overline{x}] = [4, 6]$ we obtain the "solution" $[\underline{x}, \overline{x}] = [4, 2]$, which according to Lodwick and Dubois [7] is "empty" solution set.

**Example 2.** *This example concerns the equation* $[3, 5]X = [7, 9]$, *in which, assuming that X is the interval* $[\underline{x}, \overline{x}]$, *tolerance control cannot be realized at all, because the obtained solution is an improper interval* $[\underline{x}, \overline{x}] = [2.3, 1.8]$. *Next, it will be shown what solution can be obtained with the use of MIA and the previously presented algorithm. The knowledge about the system is given by Equations (53).*

$$ax = b, \quad a \in [\underline{a}, \overline{a}] = [3, 5], \quad b \in [\underline{b}, \overline{b}] = [7, 9], \quad x = ? \tag{53}$$

*Step 1.*

$$[\underline{a}, \overline{a}] = [3, 5] \rightarrow a(\gamma_a) = 3 + 2\gamma_a, \ \gamma_a \in [0, 1] \tag{54}$$
$$[\underline{b}, \overline{b}] = [3, 6] \rightarrow b(\gamma_b) = 7 + 2\gamma_b, \ \gamma_b \in [0, 1]$$

*Step 2.*

$$ax = b \rightarrow x(\gamma_a, \gamma_b) = \frac{b(\gamma_b)}{a(\gamma_a)} = \frac{7 + 2\gamma_b}{3 + 2\gamma_a}, \ \gamma_a, \gamma_b \in [0, 1] \tag{55}$$

$$BG_X = \left\{ x(\gamma_a, \gamma_b) = \frac{7 + 2\gamma_b}{3 + 2\gamma_a} \mid \forall \gamma_a \in [0, 1], \forall \gamma_b \in [0, 1] \right\}$$

*Step 3.*

$$SP_{BG_X} = \left[ \min_{\gamma_a, \gamma_b} \frac{7 + 2\gamma_b}{3 + 2\gamma_a}, \max_{\gamma_a, \gamma_b} \frac{7 + 2\gamma_b}{3 + 2\gamma_a} \right] = [1.4, 3] = [\underline{x}, \overline{x}] = X_{poss} \tag{56}$$

*Note that* $SP_{BG_X} = [1.4, 3]$ *is a normal, proper (not improper) interval. Figure 8 shows the set of possible states of the system* $(a, b, x = b/a)$ *in the projection into the space* $A \times X$ *with the values of the third variable* $b = ax$ *plotted.*

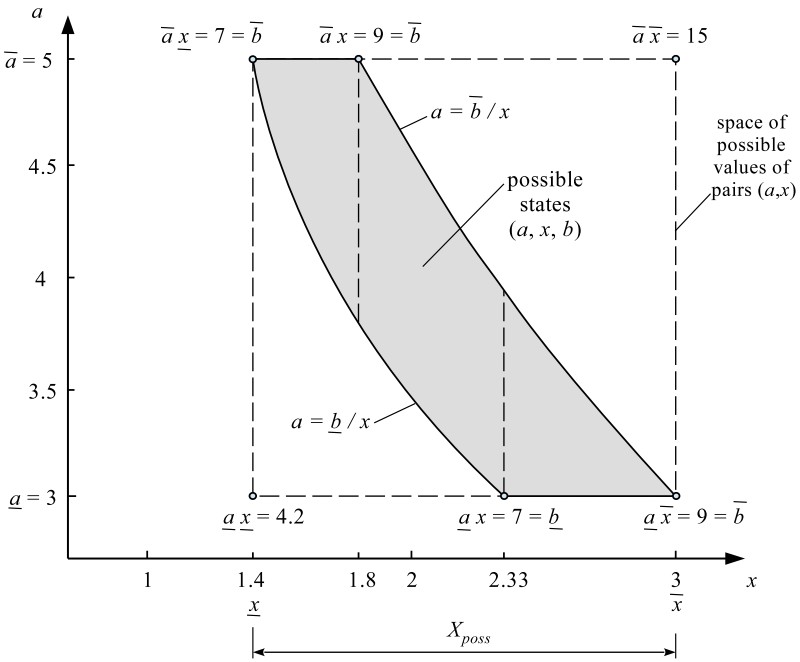

**Figure 8.** The set $S$ of possible states $(a, b, x)$ of the system $ax = b$, $a \in [3, 5]$, $b \in [7, 9]$, $x \in [1.4, 3] = X_{poss}$, in 2D-space $A \times X$.

In order to determine the $x$-value for which the robustness $r(x)$ to the disturbance $a$ is the greatest, it is enough to examine $r(x)$ at 4 characteristic points (L-lower, U-upper) $x_{LL} = \underline{b}/\underline{a} = 2.33$, $x_{LU} = \underline{b}/\bar{a} = 1.4$, $x_{UL} = \bar{b}/\underline{a} = 3$, $x_{UU} = \bar{b}/\bar{a} = 1.8$. Then we rank these values, as shown in (57).

$$x_{LU} = 1.4, \ x_{UU} = 1.8, \ x_{LL} = 2.33, \ x_{UL} = 3 \tag{57}$$

Figure 8 shows that for the outer values $x_{LU} = 1.4$, $x_{UL} = 3$ robustness $r(x) = 0$. Hence the greatest robustness can have the inner characteristic values $x_{UU} = 1.8$, $x_{LL} = 2.33$. The robustness for these $x$-values can be calculated from the Formula (58), which applies to the inner, second $x$-control zone.

$$r(x) = \frac{(\bar{b}/x) - (\underline{b}/x)}{\bar{a} - \underline{a}} \tag{58}$$

The calculation gives the results $r(x = 1.8) = 0.556$, $r(x = 2.33) = 0.429$. The distribution of the robustness for individual possible controls $x$ is shown in Figure 9.

As the distribution of the robustness $r(x)$ on Figure 9 shows, it is impossible to obtain full robustness for the disturbance change. This situation occurs in the case of many problems described by the interval equation $[\underline{a}, \bar{a}]x = [\underline{b}, \bar{b}]$. This is a frequent situation and hence realistic. Uncertainty problems where $r(x) = 1$ can be obtained are rare in practice because real problems usually have more than just 1 uncertainty generator. However, even when full robustness cannot be obtained, we should not wring our hands, because the incomplete robustness can be maximized by appropriate selection of $x$-control. As shown in Example 2, the optimal value of $x$ can be detected.

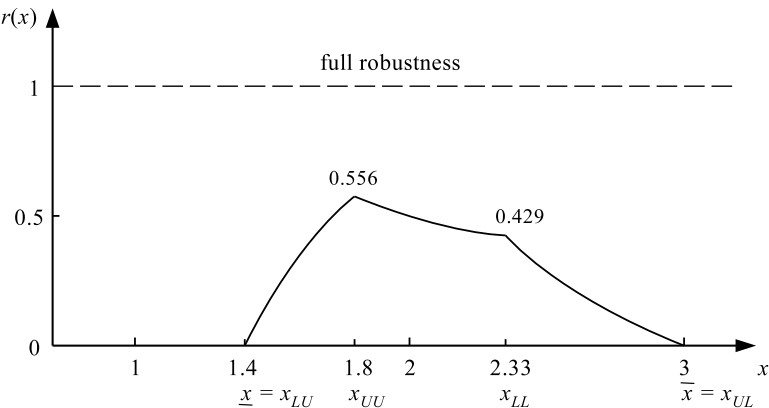

**Figure 9.** Distribution of the achievable robustness $r(x)$ for the individual possible control values $x$: optimal value $x_{opt} = 1.8$, $r(x = 1.8) = 0.556$.

In Examples 1 and 2, there are intervals $[\underline{a}, \bar{a}]$ and $[\underline{b}, \bar{b}]$ that do not contain zero. Examples with one unknown $x$ with intervals containing zero will be shown in the following. In contrast, these intervals will be represented as $[\underline{a}, 0, \bar{a}]$ and $[\underline{b}, 0, \bar{b}]$.

**Example 3.** *In this example, an equation of the form $[\underline{a}, \bar{a}]x = [\underline{b}, 0, \bar{b}]$ in which the tolerance corridor contains zero will be solved. The equation in the classic interval version (59) and in the RDM version (60):*

$$[3, 5]x = [-1, 0, 2], \tag{59}$$
$$(3 + 2\gamma_a)x = (-1 + 3\gamma_b). \tag{60}$$

*After solving the equation with the previously described method, the bag span $SP_{BG_X}$ is obtained in the form:*

$$SP_{BG_X} = [-1/3, 0, 2/3] = [\underline{x}, \bar{x}] = X_{poss}. \tag{61}$$

Figure 10 shows the set $S = \{(a,b,x)\}$ of possible states $(a,b,x)$ of the system as a projection into the space $A \times X$.

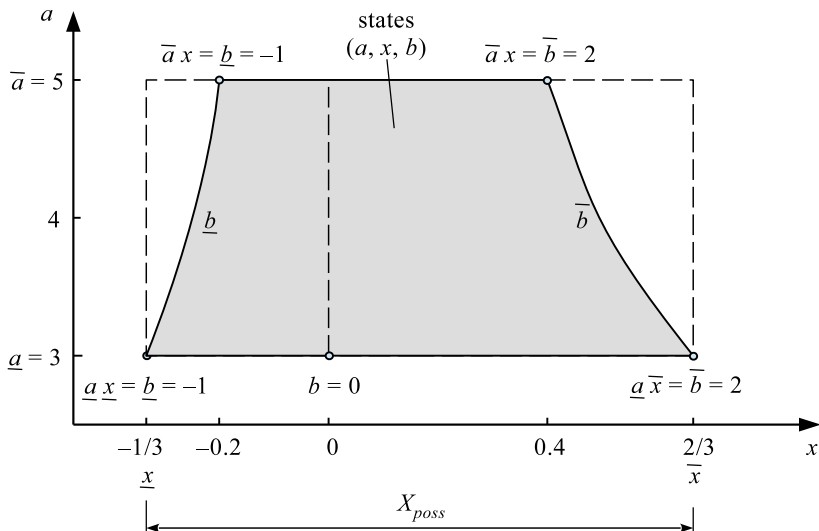

**Figure 10.** Set of possible states $S = \{(a,b,x)\}$ of the system $[3,5]x = [-1,0,2]$ in projection on the 2D-space $A \times X$ with range $X_{poss}$ of possible control values $x$.

Based on Figure 10, robustness functions can be determined for 3 ranges: $x \in [-1/3, -0.2]$, $x \in [-0.2, 0.4]$, and $x \in [0.4, 2/3]$, Formula (62).

$$
\begin{aligned}
&\text{for } x \in [-1/3, -0.2] &\qquad r(x) &= -1/x - 3 \\
&\text{for } x \in [-0.2, 0.4] &\qquad r(x) &= 1 \\
&\text{for } x \in [0.4, 2/3] &\qquad r(x) &= 2/x - 3
\end{aligned}
\tag{62}
$$

The robustness distribution of the control variable $x$ is shown in Figure 11.

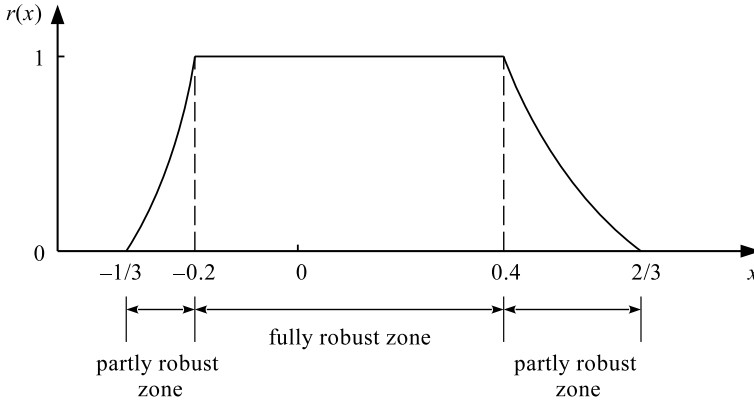

**Figure 11.** Robustness distribution $r(x)$ of the tolerance control of the system $[3,5]x = [-1,0,2]$.

As shown in Formula (62) and Figure 11, each control value $x \in [-0.2, 0, 0.4]$ allows a fully robust tolerance control to be obtained. Example 3 shows that the presence of zero in the tolerance interval $[\underline{b}, 0, \overline{b}]$ does not make it difficult to determine the set of optimal control values $x$.

**Example 4.** *System with zero in the disturbance interval $[\underline{a}, 0, \overline{a}]$. In this example, we will look for a tolerance control $x$ for a system with inputs $a$ and $x$ and output $b$ that is ruled by the multiplicative relationship $ax = b$. The knowledge about this system is given by the Formula (63).*

$$
ax = b, \quad a \in [-1, 0, 2], \quad b \in [7, 9]
\tag{63}
$$

*Formula (64) describes the system using the classical interval equation.*

$$[\underline{a}, 0, \overline{a}]x = [\underline{b}, \overline{b}] : \quad [-1, 0, 2]x = [7.9] \tag{64}$$

*As it follows from the knowledge about the input a, the state $a = 0$ is possible. This would mean the need to find the value of x that would be able to accomplish the task (65).*

$$0x = [7, 9] \tag{65}$$

*However, the state $a = 0$ means that the system operation is turned off and the task $ax \in [7, 9]$ cannot be performed. Because at $a = 0$ any control is impossible, this state should be excluded from the analysis, treating it as a special and very unlikely state. This results in the necessity to divide the set $[\underline{a}, 0, \overline{a}] = [-1, 0, 2]$ into 2 disjoint sets $a_1 \in [-a, -\Delta]$ and $a_2 \in [\Delta, a]$ that is into sets $[-1, -\Delta]$ and $[\Delta, 2]$ where $\Delta$ is a very small positive number $\Delta \approx 0$, e.g., $\Delta = 0.001$. Then the original control task (64) is decomposed into 2 tasks (66).*

$$[\underline{a}, 0, \overline{a}]x = [\underline{b}, \overline{b}] \rightarrow ([\underline{a}, -\Delta]x = [\underline{b}, \overline{b}]) \cup ([\Delta, \overline{a}]x = [\underline{b}, \overline{b}]) \tag{66}$$
$$[-1, 0, 2]x = [7, 9] \rightarrow ([-1, -\Delta]x = [7, 9]) \cup ([\Delta, 2]x = [7, 9]) \tag{67}$$

*First, the first component problem will be solved, which in terms of MIA has the form (68).*

$$[-1, -\Delta]x = [7, 9] \rightarrow (-1 + \gamma_{a1}(-\Delta + 1))x = 7 + 2\gamma_b, \quad \gamma_{a1}, \gamma_b \in [0, 1] \tag{68}$$

*The solution to this problem is given by Formula (69).*

$$BG_X = x(\gamma_{a1}, \gamma_b) = \frac{7 + 2\gamma_b}{-1 + \gamma_{a1}(-\Delta + 1)}, \quad \forall \gamma_{a1}, \forall \gamma_b \in [0, 1] \tag{69}$$

*The span of the bag $BG_X$ is determined by Formula (70).*

$$SP_{BG_X} = X_{poss} = [(-9/\Delta), -7], \quad \Delta \approx 0 \tag{70}$$

*The number $(-9/\Delta)$ in the interval (70) is a very large negative number. Figure 12 shows the set $S_1 = \{(a, x, b)\}$ of the possible states of the system for $x < 0$.*

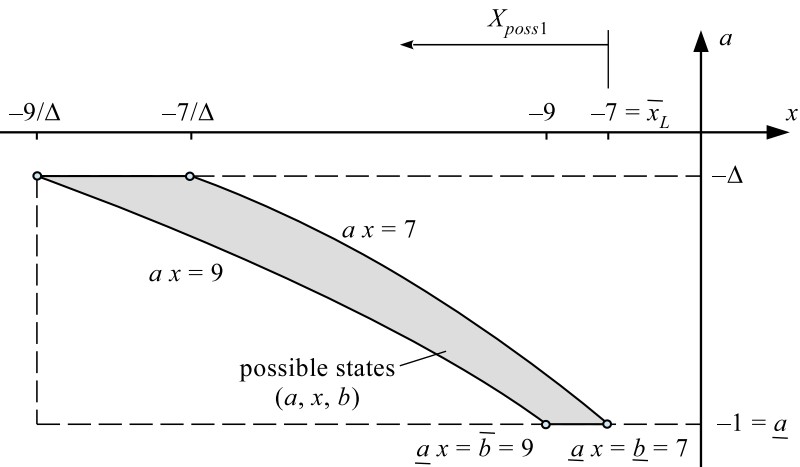

**Figure 12.** Illustrative visualization of the left set $S_1 = \{(a, x, b)\}$ of possible states $(a, x, b)$ of the system $[-1, -\Delta]x = [7, 9]$, $\Delta \approx 0$.

Figure 12 and Formula (70) show that the implementation of tolerant control in the problem under consideration for negative values of $x$ is possible only for $x \leq -7$. However, it is not possible to obtain here fully robust control but only partly robust one. Robustness function of possible controls $x$ is given by the Formula (71).

$$\text{for } x \in [-9/\Delta, -7/\Delta] \qquad r(x) = -9/(3x), \ \Delta \approx 0, \ x < 0 \tag{71}$$
$$\text{for } x \in [-7/\Delta, -9] \qquad r(x) = -2/(3x)$$
$$\text{for } x \in [-9, -7] \qquad r(x) = 1/3 + 7/(3x)$$
$$\text{for } x \in [-7, 0] \qquad r(x) = 0$$

The robustness distribution for possible negative control values $x$ is shown in Figure 13. The equation describing the system for $x > 0$ has the form (72).

$$[\Delta, 2]x = [7, 9] \to (\Delta + \gamma_{a2}(2 - \Delta))x = 7 + 2\gamma_b, \ \ \gamma_{a2}, \gamma_b \in [0, 1] \tag{72}$$

Solving Equation (72) we obtain Formula (73) for a bag of possible values $x$.

$$BG_X = x(\gamma_{a2}, \gamma_b) = \frac{7 + 2\gamma_b}{\Delta + \gamma_{a2}(2 - \Delta)}, \ \ \forall \gamma_{a2}, \forall \gamma_b \in [0, 1] \tag{73}$$

The span $SP_{BG_X}$ of the bag $BG_X$ is determined by Formula (74).

$$SP_{BG_X} = X_{poss} = \left[ \min_{\gamma_{a2}, \gamma_b} x(\gamma_{a2}, \gamma_b), \max_{\gamma_{a2}, \gamma_b} x(\gamma_{a2}, \gamma_b) \right] = [7/3, 9/\Delta], \quad \Delta \approx 0 \tag{74}$$

Figure 14 shows the set $S_2 = \{(a, x, b)\}$ of states $(a, x, b)$ of the system allowing tolerance control for $x > 0$.

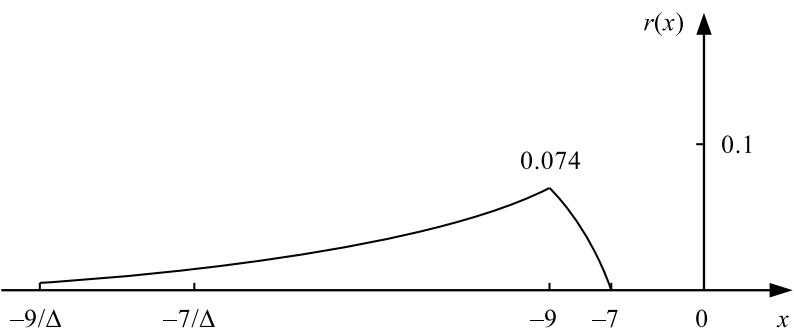

**Figure 13.** Robustness distribution $r(x)$ of possible control values $x < 0$ of the system $[-1, 0, 2]x = [7, 9]$.

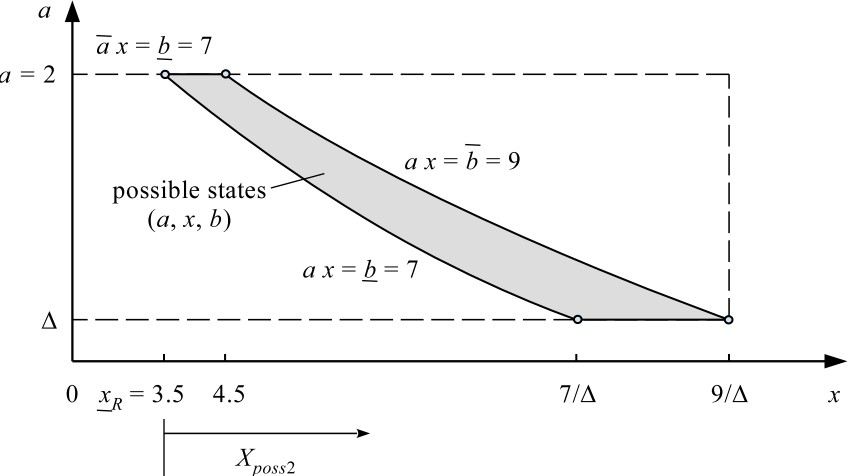

**Figure 14.** The set $S_2 = \{(a, b, x)\}$ of states $(a, b, x)$ of the system $[\Delta, 2]x = [7, 9]$ enabling tolerance control for $x > 0$.

The robustness distribution $r(x)$ of possible controls $x$, for $x > 0$, is given by Formula (75) and is shown in Figure 15.

$$
\begin{aligned}
&\text{for } x \in [0, 3.5] && r(x) = 0 && (75)\\
&\text{for } x \in [3.5, 4.5] && r(x) = 2/3 - 7/(3x)\\
&\text{for } x \in [4.5, \infty] && r(x) = 2/(3x)
\end{aligned}
$$

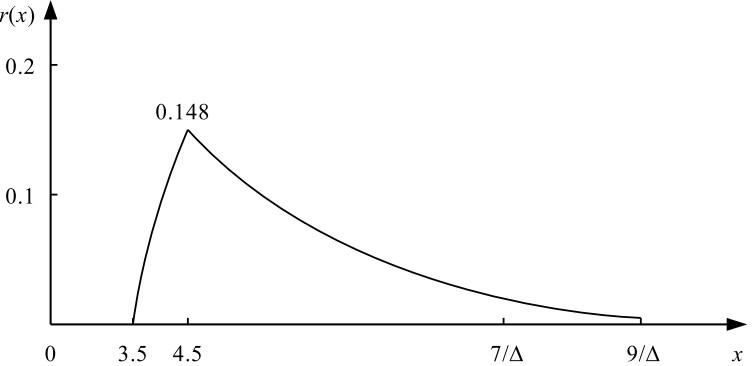

**Figure 15.** Robustness distribution $r(x)$ of the tolerance control $x$ for $x > 0$.

As shown by Formula (75) and Figure 15, the realization of the fully robust tolerance control for $x > 0$ is not possible. In this area, at most robustness $r(x) = 0.148$ can be obtained on condition that the disturbance $a$ is contained only in its upper range $a \in [\Delta, 2]$. However, the full interval of this disturbance is $[-1, 0, 2]$. In order to decide on the choice of the optimal control value $x$, it is necessary to consider the total distribution $r(x)$ presented in Figure 16.

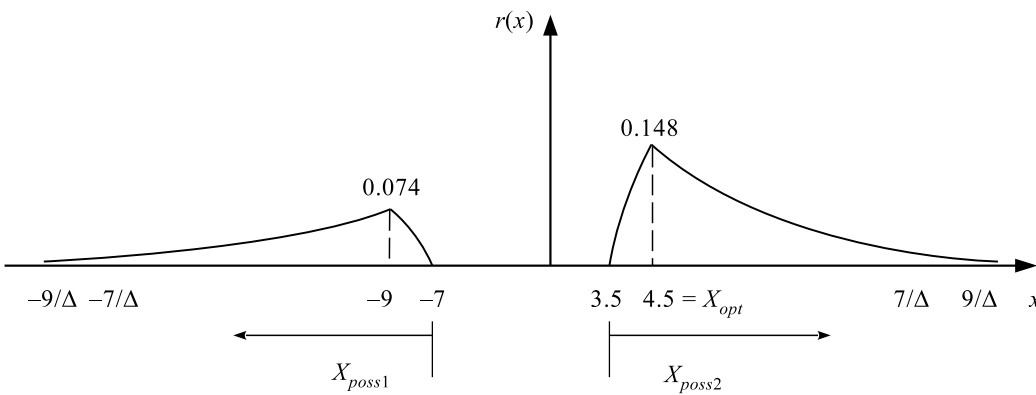

**Figure 16.** Robustness distribution $r(x)$ of control $x$ for possible changes of the disturbance $a \in [-1, 0, 2]$ of the system $[-1, 0, 2]x = [7, 9]$, together with the optimal control value $x_{opt} = 4.5$.

The interpretation of the results shown in Figure 16 is as follows. In the system under consideration, it is not possible to implement fully robust tolerance control. Only partly robust TC can be realized. PR-TC can be obtained by either negative or positive $x$ controls excluding the range $x \in [-7, 0, 3.5]$. The most preferable control is $x = 4.5$, giving the robustness of 0.148 to find $b$ in the range of tolerance corridor $b \in [7, 9]$. If disturbance $a$ were $a = 0$, then any influence on output $b$ would become impossible. However, the probability of such an event is infinitely small.

**Example 5.** *Real-life case. A doctor recommended to a 70-year old patient that his daily diet should contain [25, 40] µg of vitamin D (tolerance corridor $[\underline{b}, \overline{b}]$). To supply him with this vitamin, he recommended eating sea fish (such as: mackerel, herring) containing healthy fats. The content of vitamin D in such fish is not constant and varies in the range of [5, 10] µg for every 100 g of fish*

(uncertainty $[\underline{a}, \overline{a}]$). Neither too little nor too much vitamin D is recommended for the patient for health reasons. How much fish x [100 g] should the patient eat per day?

We want to solve the equation:

$$[\underline{a}, \overline{a}]x = [\underline{b}, \overline{b}],$$

where: $a \in [5, 10]$ µg/100 g, and $b \in [25, 40]$ µg.

$$a(\gamma_a) = 5 + 5\gamma_a, \quad b(\gamma_b) = 25 + 15\gamma_b, \quad \gamma_a, \gamma_b \in [0, 1]$$

$$x(\gamma_a, \gamma_b) = \frac{b(\gamma_b)}{a(\gamma_a)} = \frac{25 + 15\gamma_b}{5 + 5\gamma_a}$$

This task cannot be solved using the one-dimensional interval arithmetic. The set of possible states of the system $[5, 10]x = [25, 40]$ is presented in Figure 17. Solving the problem with the use of MIA we can determine the general range of possible solutions:

$$X_{poss} = [\underline{x}, \overline{x}] = \left[\min_{\gamma_a, \gamma_b} \frac{25 + 15\gamma_b}{5 + 5\gamma_a}, \max_{\gamma_a, \gamma_b} \frac{25 + 15\gamma_b}{5 + 5\gamma_a}\right] = [2.5, 8]$$

and the robustness function $r(x)$ which is presented in Figure 18.

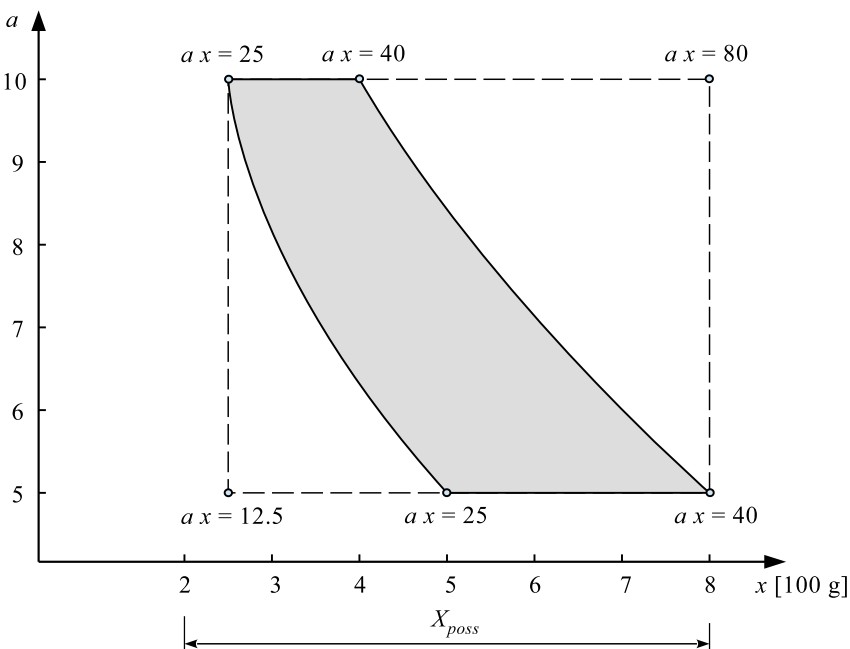

**Figure 17.** Set of possible states of the system $[5, 10]x = [25, 40]$ in projection on the 2D-space $A \times X$.

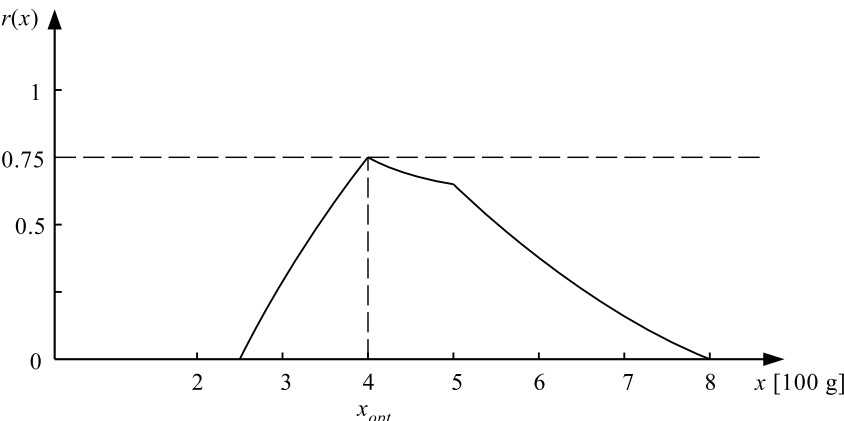

**Figure 18.** Robustness distribution $r(x)$ of the solution $x$.

Figure 18 shows that the optimal amount is $x_{opt} = 4 \times (100\,\text{g}) = 400\,\text{g}$ of fish per day. The robustness of this value to the uncertainty of vitamin D content in fish is the highest and equal to 0.75.

## 5. Conclusions

The article shows that determining the tolerant control (TC), even for uncertain static systems described by the simplest form of the interval equation $[\underline{a}, \overline{a}]X = [\underline{b}, \overline{b}]$, is not an easy task, because in many cases it is impossible to find such a value of control (decision) $x$, which will ensure a reliable hit of the $ax$ value in tolerance corridor $[\underline{b}, \overline{b}]$. However, even in the most difficult tasks it is possible to define partly robust control and it is possible to find its optimal value $x_{opt}$, which with the maximum possible robustness will give a chance to achieve TC. The reason for the inability to obtain fully robust TC is the presence of disturbance uncertainty generators (DUGs), i.e., various types of uncertain data. As their number increases, the possibility of implementing of fully robust TC decreases quickly. It is rather consistent with the intuition of engineers and scientists. In practice, the conditions for perfect implementation of the control objectives are rare. The paper explains the new approach to determining TC and indicates that it is a starting point for solving more complicated forms of static control, which are described by systems of uncertain, interval and fuzzy linear, quadratic, etc. equations. This opens the way for interesting research. The article also explains the hitherto unexplained Lodwick's anomaly concerning the interval equations.

In conclusion, the key issues presented in the paper are as follows:

- the new method based on MIA for solving the basic linear interval equation;
- the method of determination of the solution robustness;
- the method of determination of the solution that is optimal in terms of control robustness for the uncertainty of system parameters; and
- the method of determination of the optimal tolerance control for the equation $[\underline{a}, 0, \overline{a}]X = [\underline{b}, \overline{b}]$ in which the interval uncertainty on the left side of the equation contains zero.

**Author Contributions:** Conceptualization, A.P.; Formal analysis, A.P.; Investigation, A.P. and M.P.; Methodology, A.P. Software, M.P.; Visualization, M.P.; Writing—original draft, A.P.; Writing—review & editing, M.P. All authors have read and agreed to the published version of the manuscript.

**Funding:** This research received no external funding.

**Conflicts of Interest:** The authors declare no conflict of interest.

## Abbreviations

The following abbreviations are used in this manuscript:

| | |
|---|---|
| IA | interval arithmetic |
| SIA | standard interval arithmetic |
| UBM | unnatural behavior in modeling |
| LIEA | Lodwick's interval equation anomaly |
| MIA | multidimensional interval arithmetic |
| TSS | tolerance solution set |
| LTP | linear tolerance problem |
| ILS | interval linear system |
| $n$V-ILS | $n$-variable interval linear system |
| PTS | probability of tolerance satisfaction |
| DUG | disturbance uncertainty generator |
| TO | tolerance output |
| TC | tolerance control |
| PrTC | probabilistic tolerance control |
| IE | interval equation |
| FA | fuzzy arithmetic |

| TU | tolerance uncertainty |
| BG | bag |
| SP | span |
| CofG | center of gravity |

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
