# Peer review of "The Optimal Tolerance Solution of the Basic Interval Linear Equation and the Explanation of the Lodwick’s Anomaly"

_applsci, doi:10.3390/app12094382_

Round 1

Reviewer 1 Report

In this paper, by using the multidimensional Interval arithmetic, the authors investigate a new method for solving interval linear systems.

The paper is well written and organized. The provided illustrative examples and applications are interesting. However, more background on interval analysis should be included in the introduction section. Also, a comparative analysis should be discussed regarding the different types of methods introduced in the literature. In this direction, the authors are recommended the following recent papers, strongly connected with the subject studied in this paper, namely: Optimality conditions and duality for a class of generalized convex interval-valued optimization problems, Mathematics, Vol. 9, No. 22, 2979, 2021; On a class of interval-valued optimization problems, Continuum Mechanics and Thermodynamics, Vol. 34, No. 2, 617-626, 2022.

In consequence, a minor revision is required.

Reviewer 2 Report

This paper proposes a method to determine “realistic tolerance solutions” using the multidimensional interval arithmetic framework. I believe the new content belongs to the definition of “fully robust” solution set (X_FR) and “partly robust” solution set (X_PR). When evaluating a manuscript, I tend to focus on the novelty or original contribution of the manuscript. In my view, the original contribution is on the weak side, along with some technical issues.

The manuscript has five sections, where I agree that the authors have provided comprehensive backgrounds on the topic of interval arithmetic in Sections 1-3. The issue of the Lodwick’s anomaly is also well explained. I believe Section 4 represents the original contribution of the manuscript, where I guess Equations (44) to (47) are original. The idea of robustness is interesting but it is hardly considered as brand new. The robustness formulations (e.g., Equations (52) and (53)) are basically ratios for simple cases (i.e., solving one “linear” equation) with specific insight for generalization. I notice that the “explanation of the Lodwick’s anomaly” is also claimed as a contribution, but I would say that it is also weak. When I read the introduction (e.g., Figure 2), I can basically explain to myself why the solutions from M1 and M2 are different.

Besides the comment with the contribution, I would also comment that the manuscript has some technical issues (or challenges) that should be addressed. These comments are listed below.

  • I prefer the manuscript to have a more concise approach to articulate the mathematical framework. The quotation from Line 86 to Line 97 (from Shary (1995) is basically not acceptable. In addition, this “copied” background to such details seems no necessary toward the understanding of Section 4. For example, the manuscript does not demonstrate cases with a matrix A or a general procedure to solve more than one equation. I also have a similar comment to the content of the center of gravity.
  • The manuscript argues whether “ILSs are really linear”. I somehow disagree with the manuscript’s view. By the notion of “linear”, the typical interpretation is to check the output effects with respect to the inputs. With the equation aX = b, I will take X as input but ‘a’ is not an input (or a decision to be made). The uncertainty of ‘a’ does not make the output ‘b’ being non-linear with respect to X. In the manuscript’s interpretation, it seems to take ‘a’ as another input parameter, which I do not agree. Notably, analyzing the problem by varying the values of ‘a’ does not make ‘a’ be an input (or a decision variable) in the interpretation of the system.
  • As a minor comment, I somewhat disagree to interpret ‘alpha’ in the manuscript being similar to the alpha-cut in the context of fuzzy sets (somehow implied in Line 165). As I see, the notions of alpha in both contexts are different.
  • While the manuscript suggest a step-by-step procedure (page 11) for problem solving, the examples explicitly states the steps only up to Step 3. In addition, the formulations of robustness r(x) vary case-by-case (i.e., Equations (52), (53), (59), (63), (72)). It seems lacking a systematic treatment to evaluate r(x), which should be discussed before the procedure (or algorithm in the manuscript’s term).
  • I believe the robustness concept of the manuscript is based on an assumption that an interval has a uniform distribution (context of probability). Otherwise, for example, the manuscript cannot state the “probability of 0.5” on page 12. Traditionally, interval arithmetic does not make this assumption. Otherwise, interval arithmetic can be translated as a problem in probability (i.e., calculations of random variables with uniform distribution).
